# Learning Trajectory Preferences for Manipulators via Iterative Improvement

**Ashesh Jain, Brian Wojcik, Thorsten Joachims, Ashutosh Saxena**
Department of Computer Science, Cornell University.
{ashesh,bmw75,tj,asaxena}@cs.cornell.edu

## Abstract

We consider the problem of learning good trajectories for manipulation tasks. This is challenging because the criterion defining a good trajectory varies with users, tasks and environments. In this paper, we propose a co-active online learning framework for teaching robots the preferences of its users for object manipulation tasks. The key novelty of our approach lies in the type of feedback expected from the user: the human user does not need to demonstrate optimal trajectories as training data, but merely needs to iteratively provide trajectories that slightly improve over the trajectory currently proposed by the system. We argue that this co-active preference feedback can be more easily elicited from the user than demonstrations of optimal trajectories, which are often challenging and non-intuitive to provide on high degrees of freedom manipulators. Nevertheless, theoretical regret bounds of our algorithm match the asymptotic rates of optimal trajectory algorithms. We demonstrate the generalizability of our algorithm on a variety of grocery checkout tasks, for whom, the preferences were not only influenced by the object being manipulated but also by the surrounding environment.[1]

## 1  Introduction

Mobile manipulator robots have arms with high degrees of freedom (DoF), enabling them to perform household chores (e.g., PR2) or complex assembly-line tasks (e.g., Baxter). In performing these tasks, a key problem lies in identifying appropriate trajectories. An appropriate trajectory not only needs to be valid from a geometric standpoint (i.e., feasible and obstacle-free, the criterion that most path planners focus on), but it also needs to satisfy the user's preferences.

Such user's preferences over trajectories vary between users, between tasks, and between the environments the trajectory is performed in. For example, a household robot should move a glass of water in an upright position without jerks while maintaining a safe distance from nearby electronic devices. In another example, a robot checking out a kitchen knife at a grocery store should strictly move it at a safe distance from nearby humans. Furthermore, straight-line trajectories in Euclidean space may no longer be the preferred ones. For example, trajectories of heavy items should not pass over fragile items but rather move around them. These preferences are often hard to describe and anticipate without knowing where and how the robot is deployed. This makes it infeasible to manually encode (e.g. [18]) them in existing path planners (such as [29, 35]) a priori.

In this work we propose an algorithm for learning user preferences over trajectories through interactive feedback from the user in a co-active learning setting [31]. Unlike in other learning settings, where a human first demonstrates optimal trajectories for a task to the robot, our learning model does not rely on the user's ability to demonstrate optimal trajectories a priori. Instead, our learning algorithm explicitly guides the learning process and merely requires the user to incrementally improve the robot's trajectories. From these interactive improvements the robot learns a general model of the user's preferences in an online fashion. We show empirically that a small number of such interactions is sufficient to adapt a robot to a changed task. *Since the user does not have to demonstrate a (near) optimal trajectory to the robot*, we argue that our feedback is easier to provide and more widely applicable. Nevertheless, we will show that it leads to an online learning algorithm with provable regret bounds that decay at the same rate as if optimal demonstrations were available.

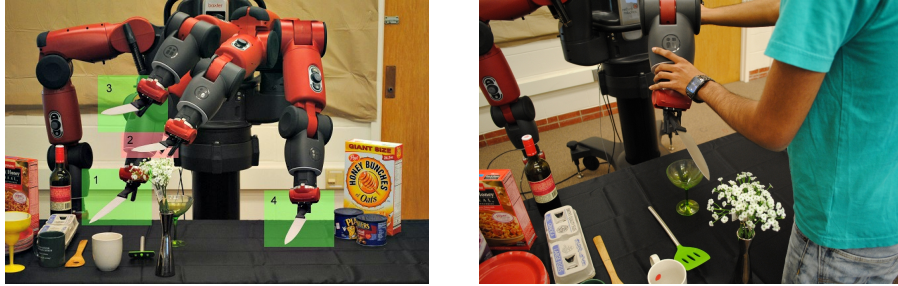

*Figure 1:* **Zero-G feedback:** Learning trajectory preferences from sub-optimal *zero-G* feedback. (**Left**) Robot plans a bad trajectory (waypoints 1-2-4) with knife close to flower. As feedback, user corrects waypoint 2 and moves it to waypoint 3. (**Right**) User providing *zero-G* feedback on waypoint 2.

In our empirical evaluation, we learn preferences for a high DoF Baxter robot on a variety of grocery checkout tasks. By designing expressive trajectory features, we show how our algorithm learns preferences from online user feedback on a broad range of tasks for which object properties are of particular importance (e.g., manipulating sharp objects with humans in vicinity). We extensively evaluate our approach on a set of 16 grocery checkout tasks, both in batch experiments as well as through robotic experiments wherein users provide their preferences on the robot. Our results show that robot trained using our algorithm not only quickly learns good trajectories on individual tasks, but also generalizes well to tasks that it has not seen before.

## 2 Related Work

Teaching a robot to produce desired motions has been a long standing goal and several approaches have been studied. Most of the past research has focused on mimicking expert's demonstrations, for example, autonomous helicopter flights [1], ball-in-a-cup experiment [17], planning 2-D paths [27, 25, 26], etc. Such a setting (learning from demonstration, LfD) is applicable to scenarios when it is clear to an expert what constitutes a good trajectory. In many scenarios, especially involving high DoF manipulators, this is extremely challenging to do [2].[2] This is because the users have to give not only the end-effector's location at each time-step, but also the full configuration of the arm in a way that is spatially and temporally consistent. In our setting, the user never discloses the optimal trajectory (or provide optimal feedback) to the robot, but instead, the robot learns preferences from sub-optimal suggestions on how the trajectory can be improved.

Some later works in LfD provided ways for handling noisy demonstrations, under the assumption that demonstrations are either near optimal [39] or locally optimal [22]. Providing noisy demonstrations is different from providing relative preferences, which are biased and can be far from optimal. We compare with an algorithm for noisy LfD learning in our experiments. A recent work [37] leverages user feedback to learn rewards of a Markov decision process. Our approach advances over [37] and Calinon et. al. [5] in that it models sub-optimality in user feedback and theoretically converges to user's hidden score function. We also capture the necessary contextual information for household and assembly-line robots, while such context is absent in [5, 37]. Our application scenario of learning trajectories for high DoF manipulations performing tasks in presence of different objects and environmental constraints goes beyond the application scenarios that previous works have considered. We design appropriate features that consider robot configurations, object-object relations, and temporal behavior, and use them to learn a score function representing the preferences in trajectories.

User preferences have been studied in the field of human-robot interaction. Sisbot et. al. [34, 33] and Mainprice et. al. [23] planned trajectories satisfying user specified preferences in form of constraints on the distance of robot from user, the visibility of robot and the user arm comfort. Dragan et. al. [8] used functional gradients [29] to optimize for legibility of robot trajectories. We differ from these in that we *learn* score functions reflecting user preferences from implicit feedback.

## 3 Learning and Feedback Model

We model the learning problem in the following way. For a given task, the robot is given a context $x$ that describes the environment, the objects, and any other input relevant to the problem. The robot has to figure out what is a good trajectory $y$ for this context. Formally, we assume that the user has a scoring function $s^*(x, y)$ that reflects how much he values each trajectory $y$ for context $x$. The higher the score, the better the trajectory. Note that this scoring function cannot be observed directly, nor do we assume that the user can actually provide cardinal valuations according to this

function. Instead, we merely assume that the user can provide us with *preferences* that reflect this scoring function. The robots goal is to learn a function $s(x, y; w)$ (where $w$ are the parameters to be learned) that approximates the users true scoring function $s^*(x, y)$ as closely as possible.

**Interaction Model.** The learning process proceeds through the following repeated cycle of interactions between robot and user.

**Step 1:** The robot receives a context $x$. It then uses a planner to sample a set of trajectories, and ranks them according to its current approximate scoring function $s(x, y; w)$.

**Step 2:** The user either lets the robot execute the top-ranked trajectory, or corrects the robot by providing an improved trajectory $\bar{y}$. This provides feedback indicating that $s^*(x, \bar{y}) > s^*(x, y)$.

**Step 3:** The robot now updates the parameter $w$ of $s(x, y; w)$ based on this preference feedback and returns to step 1.

**Regret.** The robot's performance will be measured in terms of regret, $REG_T = \frac{1}{T} \sum_{t=1}^{T} [s^*(x_t, y_t^*) - s^*(x_t, y_t)]$, which compares the robot's trajectory $y_t$ at each time step $t$ against the optimal trajectory $y_t^*$ maximizing the user's unknown scoring function $s^*(x, y)$, $y_t^* = argmax_y s^*(x_t, y)$. Note that the regret is expressed in terms of the user's true scoring function $s^*$, even though this function is *never observed*. Regret characterizes the performance of the robot over its whole lifetime, therefore reflecting how well it performs *throughout* the learning process. As we will show in the following sections, we employ learning algorithms with theoretical bounds on the regret for scoring functions that are linear in their parameters, making only minimal assumptions about the difference in score between $s^*(x, \bar{y})$ and $s^*(x, y)$ in Step 2 of the learning process.

**User Feedback and Trajectory Visualization.** Since the ability to easily give preference feedback in Step 2 is crucial for making the robot learning system easy to use for humans, we designed two feedback mechanisms that enable the user to easily provide improved trajectories.

*(a) Re-ranking:* We rank trajectories in order of their current predicted scores and visualize the ranking using OpenRave [7]. User observers trajectories sequentially and clicks on the first trajectory which is better than the top ranked trajectory.

*(b) Zero-G:* This feedback allow users to improve trajectory waypoints by physically changing the robot's arm configuration as shown in Figure 1. To enable effortless steering of robot's arm to desired configuration we leverage Baxter's zero-force gravity-compensation mode. Hence we refer this feedback as *zero-G*. This feedback is useful (i) for bootstrapping the robot, (ii) for avoiding local maxima where the top trajectories in the ranked list are all bad but ordered correctly, and (iii) when the user is satisfied with the top ranked trajectory except for minor errors. A counterpart of this feedback is keyframe based LfD [2] where an expert demonstrates a sequence of optimal waypoints instead of the complete trajectory.

Note that in both re-ranking and zero-G feedback, the user never reveals the optimal trajectory to the algorithm but just provides a slightly improved trajectory.

## 4 Learning Algorithm

For each task, we model the user's scoring function $s^*(x, y)$ with the following parameterized family of functions.

$$s(x, y; w) = w \cdot \phi(x, y) \tag{1}$$

$w$ is a weight vector that needs to be learned, and $\phi(\cdot)$ are features describing trajectory $y$ for context $x$. We further decompose the score function in two parts, one only concerned with the objects the trajectory is interacting with, and the other with the object being manipulated and the environment.

$$s(x, y; w_O, w_E) = s_O(x, y; w_O) + s_E(x, y; w_E) = w_O \cdot \phi_O(x, y) + w_E \cdot \phi_E(x, y) \tag{2}$$

We now describe the features for the two terms, $\phi_O(\cdot)$ and $\phi_E(\cdot)$ in the following.

### 4.1 Features Describing Object-Object Interactions

This feature captures the interaction between objects in the environment with the object being manipulated. We enumerate waypoints of trajectory $y$ as $y_1, .., y_N$ and objects in the environment as $\mathcal{O} = \{o_1, .., o_K\}$. The robot manipulates the object $\bar{o} \in \mathcal{O}$. A few of the trajectory waypoints would be affected by the other objects in the environment. For example in Figure 2, $o_1$ and $o_2$ affect the waypoint $y_3$ because of proximity. Specifically, we connect an object $o_k$ to a trajectory waypoint if the minimum distance to collision is less than a threshold or if $o_k$ lies below $\bar{o}$. The edge connecting $y_j$ and $o_k$ is denoted as $(y_j, o_k) \in \mathcal{E}$.

Since it is the attributes [19] of the object that really matter in determining the trajectory quality, we represent each object with its *attributes*. Specifically, for every object $o_k$, we consider a vector of $M$ binary variables $[l_k^1, .., l_k^M]$, with each $l_k^m = \{0, 1\}$ indicating whether object $o_k$ possesses

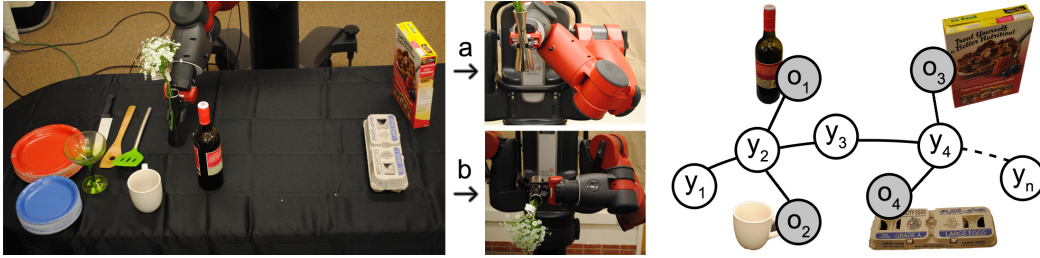

*Figure 2:* **(Left)** A grocery checkout environment with a few objects where the robot was asked to checkout flowervase on the left to the right. **(Middle)** There are two ways of moving it, 'a' and 'b', both are sub-optimal in that the arm is contorted in 'a' but it tilts the vase in 'b'. Given such constrained scenarios, we need to reason about such subtle preferences. **(Right)** We encode preferences concerned with object-object interactions in a score function expressed over a graph. Here $y_1, \ldots, y_n$ are different waypoints in a trajectory. The shaded nodes corresponds to environment (table node not shown here). Edges denotes interaction between nodes.

property $m$ or not. For example, if the set of possible properties are {heavy, fragile, sharp, hot, liquid, electronic}, then a laptop and a glass table can have labels $[0,1,0,0,0,1]$ and $[0,1,0,0,0,0]$ respectively. The binary variables $l_k^p$ and $l^q$ indicates whether $o_k$ and $\bar{o}$ possess property $p$ and $q$ respectively.[3] Then, for every $(y_j, o_k)$ edge, we extract following four features $\phi_{oo}(y_j, o_k)$: projection of minimum distance to collision along x, y and z (vertical) axis and a binary variable, that is 1, if $o_k$ lies vertically below $\bar{o}$, 0 otherwise.

We now define the score $s_O(\cdot)$ over this graph as follows:

$$s_O(x, y; w_O) = \sum_{(y_j, o_k) \in \mathcal{E}} \sum_{p,q=1}^{M} l_k^p l_k^q [w_{pq} \cdot \phi_{oo}(y_j, o_k)] \tag{3}$$

Here, the weight vector $w_{pq}$ captures interaction between objects with properties $p$ and $q$. We obtain $w_O$ in eq. (2) by concatenating vectors $w_{pq}$. More formally, if the vector at position $i$ of $w_O$ is $w_{uv}$ then the vector corresponding to position $i$ of $\phi_O(x, y)$ will be $\sum_{(y_j, o_k) \in \mathcal{E}} l_k^u l_k^v [\phi_{oo}(y_j, o_k)]$.

### 4.2 Trajectory Features

We now describe features, $\phi_E(x, y)$, obtained by performing operations on a set of waypoints. They comprise the following three types of the features:

**Robot Arm Configurations.** While a robot can reach the same operational space configuration for its wrist with different configurations of the arm, not all of them are preferred [38]. For example, the contorted way of holding the flowervase shown in Figure 2 may be fine at that time instant, but would present problems if our goal is to perform an activity with it, e.g. packing it after checkout. Furthermore, humans like to anticipate robots move and to gain users' confidence, robot should produce predictable and legible robot motion [8].

We compute features capturing robot's arm configuration using the location of its elbow and wrist, w.r.t. to its shoulder, in cylindrical coordinate system, $(r, \theta, z)$. We divide a trajectory into three parts in time and compute 9 features for each of the parts. These features encode the maximum and minimum $r$, $\theta$ and $z$ values for wrist and elbow in that part of the trajectory, giving us 6 features. Since at the limits of the manipulator configuration, joint locks may happen, therefore we also add 3 features for the location of robot's elbow whenever the end-effector attains its maximum $r$, $\theta$ and $z$ values respectively. Therefore obtaining $\phi_{robot}(\cdot) \in \mathbb{R}^9$ (3+3+3=9) features for each one-third part and $\phi_{robot}(\cdot) \in \mathbb{R}^{27}$ for the complete trajectory.

**Orientation and Temporal Behavior of the Object to be Manipulated.** Object orientation during the trajectory is crucial in deciding its quality. For some tasks, the orientation must be strictly maintained (e.g., moving a cup full of coffee); and for some others, it may be necessary to change it in a particular fashion (e.g., pouring activity). Different parts of the trajectory may have different requirements over time. For example, in the placing task, we may need to bring the object closer to obstacles and be more careful.

We therefore divide trajectory into three parts in time. For each part we store the cosine of the object's maximum deviation, along the vertical axis, from its final orientation at the goal location. To capture object's oscillation along trajectory, we obtain a spectrogram for each one-third part for

the movement of the object in $x$, $y$, $z$ directions as well as for the deviation along vertical axis (e.g. Figure 3). We then compute the average power spectral density in the low and high frequency part as eight additional features for each. This gives us 9 (=1+4*2) features for each one-third part. Together with one additional feature of object's maximum deviation along the whole trajectory, we get $\phi_{obj}(\cdot) \in \mathbb{R}^{28}$ (=9*3+1).

**Object-Environment Interactions.** This feature captures temporal variation of vertical and horizontal distances of the object $\bar{o}$ from its surrounding surfaces. In detail, we divide the trajectory into three equal parts, and for each part we compute object's: (i) minimum vertical distance from the nearest surface below it. (ii) minimum horizontal distance from the surrounding surfaces; and (iii) minimum distance from the table, on which the task is being performed, and (iv) minimum distance from the goal location. We also take an average, over all the waypoints, of the horizontal and vertical distances between the object and the nearest surfaces around it.[4] To capture temporal variation of object's distance from its surrounding we plot a time-frequency spectrogram of the object's vertical distance from the nearest surface below it, from which we extract six features by dividing it into grids. This feature is expressive enough to differentiate whether

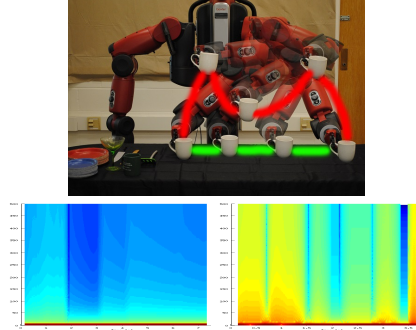

*Figure 3:* (**Top**) A good and bad trajectory for moving a mug. The bad trajectory undergoes ups-and-downs. (**Bottom**) Spectrograms for movement in z-direction: (**Right**) Good trajectory, (**Left**) Bad trajectory.

an object just grazes over table's edge (steep change in vertical distance) versus, it first goes up and over the table and then moves down (relatively smoother change). Thus, the features obtained from object-environment interaction are $\phi_{obj-env}(\cdot) \in \mathbb{R}^{20}$ (3*4+2+6=20).

Final feature vector is obtained by concatenating $\phi_{obj-env}$, $\phi_{obj}$ and $\phi_{robot}$, giving us $\phi_E(\cdot) \in \mathbb{R}^{75}$.

### 4.3 Computing Trajectory Rankings

For obtaining the top trajectory (or a top few) for a given task with context $x$, we would like to maximize the current scoring function $s(x, y; w_O, w_E)$.

$$y^* = \arg\max_y s(x, y; w_O, w_E). \tag{4}$$

Note that this poses two challenges. First, trajectory space is continuous and needs to be discretized to maintain argmax in (4) tractable. Second, for a given set $\{y^{(1)}, \ldots, y^{(n)}\}$ of discrete trajectories, we need to compute (4). Fortunately, the latter problem is easy to solve and simply amounts to sorting the trajectories by their trajectory scores $s(x, y^{(i)}; w_O, w_E)$. Two effective ways of solving the former problem is either discretizing the robot's configuration space or directly sampling trajectories from the continuous space. Previously both approaches [3, 4, 6, 36] have been studied. However, for high DoF manipulators sampling based approaches [4, 6] maintains tractability of the problem, hence we take this approach. More precisely, similar to Berg et al. [4], we sample trajectories using rapidly-exploring random tree (RRT) [20].[5] Since our primary goal is to learn a score function on sampled set of trajectories we now describe our learning algorithm and for more literature on sampling trajectories we refer the readers to [9].

### 4.4 Learning the Scoring Function

The goal is to learn the parameters $w_O$ and $w_E$ of the scoring function $s(x, y; w_O, w_E)$ so that it can be used to rank trajectories according to the user's preferences. To do so, we adapt the Preference Perceptron algorithm [31] as detailed in Algorithm 1. We call this algorithm the Trajectory Preference Perceptron (TPP). Given a context $x_t$, the top-ranked trajectory $y_t$ under the current parameters $w_O$ and $w_E$, and the user's feedback trajectory $\bar{y}_t$, the TPP updates the weights in the direction $\phi_O(x_t, \bar{y}_t) - \phi_O(x_t, y_t)$ and $\phi_E(x_t, \bar{y}_t) - \phi_E(x_t, y_t)$ respectively.

Despite its simplicity and even though the algorithm typically does not receive the optimal trajectory $y_t^* = \arg\max_y s^*(x_t, y)$ as feedback, the TPP enjoys guarantees on the regret [31]. We merely need to characterize by how much the feedback improves on the presented ranking using the following definition of expected $\alpha$-informative feedback: $E_t[s^*(x_t, \bar{y}_t)] \geq s^*(x_t, y_t) +$

$\alpha(s^*(x_t, y_t^*) - s^*(x_t, y_t)) - \xi_t$. This definition states that the user feedback should have a score of $\bar{y}_t$ that is—in expectation over the users choices—higher than that of $y_t$ by a fraction $\alpha \in (0, 1]$ of the maximum possible range $s^*(x_t, \bar{y}_t) - s^*(x_t, y_t)$. If this condition is not fulfilled due to bias in the feedback, the slack variable $\xi_t$ captures the amount of violation. In this way any feedback can be described by an appropriate combination of $\alpha$ and $\xi_t$. Using these two parameters, the proof by [31] can be adapted to show that the expected average regret of the TPP is upper bounded by $E[REG_T] \leq \mathcal{O}(\frac{1}{\alpha\sqrt{T}} + \frac{1}{\alpha T}\sum_{t=1}^{T} \xi_t)$ after $T$ rounds of feedback.

## 5 Experiments and Results

We now describe our data set, baseline algorithms and the evaluation metrics we use. Following this, we present quantitative results (Section 5.2) and report robotic experiments on Baxter (Section 5.3).

### 5.1 Experimental Setup

**Task and Activity Set for Evaluation.** We evaluate our approach on 16 pick-and-place robotic tasks in a grocery store checkout setting. To assess generalizability of our

---

**Algorithm 1** Trajectory Preference Perceptron. (TPP)

Initialize $w_O^{(1)} \leftarrow 0$, $w_E^{(1)} \leftarrow 0$
**for** $t = 1$ to $T$ **do**
    Sample trajectories $\{y^{(1)}, ..., y^{(n)}\}$
    $y_t = argmax_y s(x_t, y; w_O^{(t)}, w_E^{(t)})$
    Obtain user feedback $\bar{y}_t$
    $w_O^{(t+1)} \leftarrow w_O^{(t)} + \phi_O(x_t, \bar{y}_t) - \phi_O(x_t, y_t)$
    $w_E^{(t+1)} \leftarrow w_E^{(t)} + \phi_E(x_t, \bar{y}_t) - \phi_E(x_t, y_t)$
**end for**

---

approach, for each task we train and test on scenarios with different objects being manipulated, and/or with a different environment. We evaluate the quality of trajectories after the robot has grasped the items and while it moves them for checkout. Our work complements previous works on grasping items [30, 21], pick and place tasks [11], and detecting bar code for grocery checkout [16]. We consider following three commonly occurring activities in a grocery store:

*1) Manipulation centric:* These activities primarily care for the object being manipulated. Hence the object's properties and the way robot moves it in the environment is more relevant. Examples include moving common objects like cereal box, Figure 4 (left), or moving fruits and vegetables, which can be damaged when dropped/pushed into other items.
*2) Environment centric:* These activities also care for the interactions of the object being manipulated with the surrounding objects. Our object-object interaction features allow the algorithm to learn preferences on trajectories for moving fragile objects like glasses and egg cartons, Figure 4 (middle).
*3) Human centric:* Sudden movements by the robot put the human in a danger of getting hurt. We consider activities where a robot manipulates sharp objects, e.g., moving a knife with a human in vicinity as shown in Figure 4 (right). In previous work, such relations were considered in the context of scene understanding [10, 12].

**Baseline algorithms.** We evaluate the algorithms that learn preferences from online feedback, under two settings: (a) *untrained*, where the algorithms learn preferences for the new task from scratch without observing any previous feedback; (b) *pre-trained*, where the algorithms are pre-trained on other similar tasks, and then adapt to the new task. We compare the following algorithms:

- *Geometric*: It plans a path, independent of the task, using a BiRRT [20] planner.
- *Manual*: It plans a path following certain manually coded preferences.
- *TPP*: This is our algorithm. We evaluate it under both, *untrained* and *pre-trained* settings.
- *Oracle-svm*: This algorithm leverages the expert's labels on trajectories (hence the name *Oracle*) and is trained using SVM-rank [13] in a batch manner. This algorithm is *not realizable in practice*, as it requires labeling on the large space of trajectories. We use this only in pre-trained setting and during prediction it just predicts once and does not learn further.
- *MMP-online*: This is an online implementation of Maximum margin planning (MMP) [26, 28] algorithm. MMP attempts to make an expert's trajectory better than any other trajectory by a

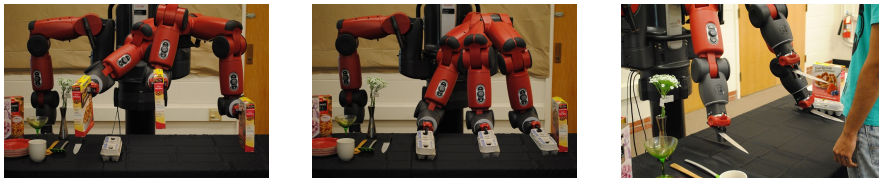

*Figure 4:* (**Left**) *Manipulation centric:* a box of cornflakes doesn't interact much with surrounding items and is indifferent to orientation. (**Middle**) *Environment centric:* an egg carton is fragile and should preferably be kept upright and closer to a supporting surface. (**Right**) *Human centric:* a knife is sharp and interacts with nearby soft items and humans. It should strictly be kept at a safe distance from humans.

margin, and can be interpreted as a special case of our algorithm with 1-informative feedback. However, adapting MMP to our experiments poses two challenges: (i) we do not have knowledge of optimal trajectory; and (ii) the state space of the manipulator we consider is too large, and discretizing makes learning via MMP intractable. We therefore train MMP from online user feedback observed on a set of trajectories. We further treat the observed feedback as optimal. At every iteration we train a structural support vector machine (SSVM) [14] using all previous feedback as training examples, and use the learned weights to predict trajectory scores for the next iteration. Since we learn on a set of trajectories, the argmax operation in SSVM remains tractable. We quantify closeness of trajectories by the $l_2-$norm of difference in their feature representations, and choose the regularization parameter $C$ for training SSVM in hindsight, to give an unfair advantage to MMP-online.

**Evaluation metrics.** In addition to performing a user study on Baxter robot (Section 5.3), we also designed a data set to quantitatively evaluate the performance of our online algorithm. An expert labeled 1300 trajectories on a Likert scale of 1-5 (where 5 is the best) on the basis of subjective human preferences. Note that these absolute ratings are never provided to our algorithms and are only used for the quantitative evaluation of different algorithms. We quantify the quality of a ranked list of trajectories by its normalized discounted cumulative gain (nDCG) [24] at positions 1 and 3. While nDCG@1 is a suitable metric for autonomous robots that execute the top ranked trajectory, nDCG@3 is suitable for scenarios where the robot is supervised by humans.

## 5.2  Results and Discussion

We now present the quantitative results on the data set of 1300 labeled trajectories.

**How well does TPP generalize to new tasks?** To study generalization of preference feedback we evaluate performance of TPP-pre-trained (i.e., *TPP* algorithm under *pre-trained* setting) on a set of tasks the algorithm has not seen before. We study generalization when: (a) only the object being manipulated changes, e.g., an egg carton replaced by tomatoes, (b) only the surrounding environment changes, e.g., rearranging objects in the environment or changing the start location of tasks, and (c) when both change. Figure 5 shows nDCG@3 plots averaged over tasks for all types of activities.[6] TPP-pre-trained starts-off with higher nDCG@3 values than TPP-untrained in all three cases. Further, as more feedback is received, performance of both algorithms improve to eventually become (almost) identical. We further observe, generalizing to tasks with both new environment and object is harder than when only one of them changes.

**How does TPP compare to other algorithms?** Despite the fact that TPP never observes optimal feedback, it performs better than baseline algorithms, see Figure 5. It improves over Oracle-SVM in less than 5 feedbacks, which is not updated since it requires expert's labels on test set and hence it is impractical. MMP-online assumes every user feedback as optimal, and over iterations

*Table 1:* **Comparison of different algorithms and study of features in untrained setting.** Table contains average nDCG@1(nDCG@3) values over 20 rounds of feedback.

|  | Algorithms | Manipulation centric | Environment centric | Human centric | Mean |
|---|---|---|---|---|---|
|  | Geometric | 0.46 (0.48) | 0.45 (0.39) | 0.31 (0.30) | 0.40 (0.39) |
|  | Manual | 0.61 (0.62) | 0.77 (0.77) | 0.33 (0.31) | 0.57 (0.57) |
| TPP Features | Obj-obj interaction | 0.68 (0.68) | 0.80 (0.79) | 0.79 (0.73) | 0.76 (0.74) |
| TPP Features | Robot arm config | 0.82 (0.77) | 0.78 (0.72) | 0.80 (0.69) | 0.80 (0.73) |
| TPP Features | Object trajectory | 0.85 (0.81) | 0.88 (0.84) | 0.85 (0.72) | 0.86 (0.79) |
| TPP Features | Object environment | 0.70 (0.69) | 0.75 (0.74) | 0.81 (0.65) | 0.75 (0.69) |
|  | TPP (all features) | **0.88 (0.84)** | **0.90 (0.85)** | **0.90 (0.80)** | **0.89 (0.83)** |
|  | MMP-online | 0.47 (0.50) | 0.54 (0.56) | 0.33 (0.30) | 0.45 (0.46) |

accumulates many contradictory training examples. This also highlights the sensitivity of MMP to sub-optimal demonstrations. We also compare against planners with manually coded preferences e.g., keep a flowervase upright. However, some preferences are difficult to specify, e.g., not to move heavy objects over fragile items. We empirically found the resulting manual algorithm produces poor trajectories with an average nDCG@3 of 0.57 over all types of activities.

**How helpful are different features?** Table 1 shows the performance of the TPP algorithm in the untrained setting using different features. Individually each feature captures several aspects indicating goodness of trajectories, and combined together they give the best performance. Object trajectory features capture preferences related to the orientation of the object. Robot arm configuration and object environment features capture preferences by detecting undesirable contorted arm configurations and maintaining safe distance from surrounding surfaces, respectively. Object-object features by themselves can only learn, for example, to move egg carton closer to a supporting surface, but might still move it with jerks or contorted arms. These features can be combined with other features to yield more expressive features. Nevertheless, by themselves they perform better than Manual algorithm. Table 1 also compares TPP and MMP-online under untrained setting.

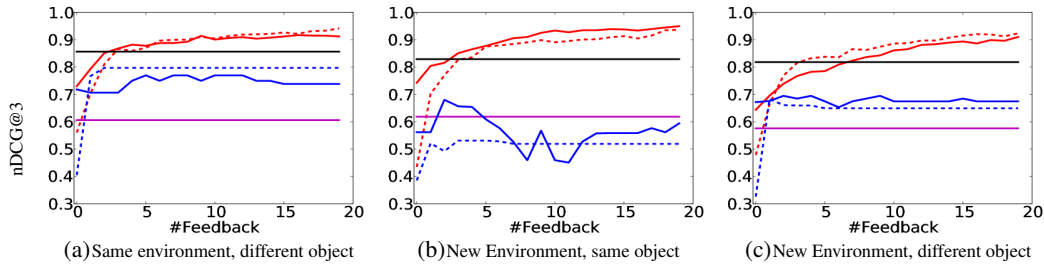

(a)Same environment, different object    (b)New Environment, same object    (c)New Environment, different object

*Figure 5:* Study of generalization with change in object, environment and both. Manual, Oracle-SVM, Pretrained MMP-online (—), Untrained MMP-online (– –), Pre-trained TPP (—), Untrained TPP (– –).

## 5.3 Robotic Experiment: User Study in learning trajectories

We perform a user study of our system on Baxter robot on a variety of tasks of varying difficulties. Thereby, showing our approach is practically realizable, and that the combination of re-rank and zero-G feedbacks allows the users to train the robot in few feedbacks.

**Experiment setup:** In this study, five users (not associated with this work) used our system to train Baxter for grocery checkout tasks, using zero-G and re-rank feedback. Zero-G was provided kinesthetically on the robot, while re-rank was elicited in a simulator (on a desktop computer). A set of 10 tasks of varying difficulty level was presented to users one at a time, and they were instructed to provide feedback until they were satisfied with the top ranked trajectory. To quantify the quality of learning each user evaluated their own trajectories (self score), the trajectories learned of the other users (cross score), and those predicted by Oracle-svm, on a Likert scale of 1-5 (where 5 is the best). We also recorded the time a user took for each task—from start of training till the user was satisfied.

**Results from user study.** The study shows each user on an average took 3 rerank and 2 zero-G feedbacks to train Baxter (Table 2). Within 5 feedbacks the users were able to improve over Oracle-svm, Fig. 6 (Left), consistent with our previous analysis. Re-rank feedback was popular for easier tasks, Fig. 6 (Right). However as difficulty increased the users relied more on zero-G feedback, which allows rectifying erroneous waypoints precisely. An average difference of 0.6 between users' self and cross score suggests preferences marginally varied across the users.

*Table 2:* Shows learning statistics for each user averaged over all tasks. The number in parentheses is standard deviation.

| User | # Re-ranking feedback | # Zero-G feedback | Average time (min.) | Trajectory Quality self | cross |
|---|---|---|---|---|---|
| 1 | 5.4 (4.1) | 3.3 (3.4) | 7.8 (4.9) | 3.8 (0.6) | 4.0 (1.4) |
| 2 | 1.8 (1.0) | 1.7 (1.3) | 4.6 (1.7) | 4.3 (1.2) | 3.6 (1.2) |
| 3 | 2.9 (0.8) | 2.0 (2.0) | 5.0 (2.9) | 4.4 (0.7) | 3.2 (1.2) |
| 4 | 3.2 (2.0) | 1.5 (0.9) | 5.3 (1.9) | 3.0 (1.2) | 3.7 (1.0) |
| 5 | 3.6 (1.0) | 1.9 (2.1) | 5.0 (2.3) | 3.5 (1.3) | 3.3 (0.6) |

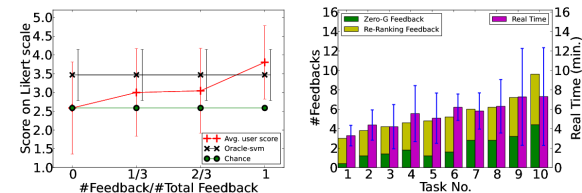

In terms of training time, each user took on average 5.5 minutes per-task, which we believe is acceptable for most applications. Future research in human computer interaction, visualization and better user interface [32] could further reduce this time.

*Figure 6:* **(Left)** Average quality of the learned trajectory after every one-third of total feedback. **(Right)** Bar chart showing the average number of feedback and time required for each task. Task difficulty increases from 1 to 10.

Despite its limited size, through user study we show our algorithm is realizable in practice on high DoF manipulators. We hope this motivates researchers to build robotic systems capable of learning from *non-expert* users.

For more details and video, please visit: `http://pr.cs.cornell.edu/coactive`

## 6 Conclusion

In this paper we presented a co-active learning framework for training robots to select trajectories that obey a user's preferences. Unlike in standard learning from demonstration approaches, our framework does not require the user to provide optimal trajectories as training data, but can learn from iterative improvements. Despite only requiring weak feedback, our TPP learning algorithm has provable regret bounds and empirically performs well. In particular, we propose a set of trajectory features for which the TPP generalizes well on tasks which the robot has not seen before. In addition to the batch experiments, robotic experiments confirmed that incremental feedback generation is indeed feasible and that it leads to good learning results already after only a few iterations.

**Acknowledgments.** We thank Shikhar Sharma for help with the experiments. This research was supported by ARO, Microsoft Faculty fellowship and NSF Career award (to Saxena).

## Footnotes

[1]For more details and a demonstration video, visit: `http://pr.cs.cornell.edu/coactive`

[2]Consider the following analogy: In search engine results, it is much harder for a user to provide the best web-pages for each query, but it is easier to provide relative ranking on the search results by clicking.

[3]In this work, our goal is to relax the assumption of unbiased and close to optimal feedback. We therefore assume complete knowledge of the environment for our algorithm, and for the algorithms we compare against. In practice, such knowledge can be extracted using an object attribute labeling algorithm such as in [19].

[4]We query PQP collision checker plugin of OpenRave for these distances.

[5]When RRT becomes too slow, we switch to a more efficient bidirectional-RRT. The cost function (or its approximation) we learn can be fed to trajectory optimizers like CHOMP [29] or optimal planners like RRT* [15] to produce reasonably good trajectories.

[6] Similar results were obtained with nDCG@1 metric. We have not included it due to space constraints.

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
