[Reviews · NeurIPS 2013]

Submitted by Assigned_Reviewer_1

The paper present an co-active learning approach to generate user- or task-preferred trajectories for manipulation accounting for task-, object-, environment- and human-related context features.

The paper addresses a timely topic with an interesting approach based on the rather novel idea of incorporating task context into the problem of trajectory learning. It is understandable and easy to read, and presents an evaluation with quantitative and subjective measures, the former carried out in simulation with a 7-dof PR2 arm

However, there are weaknesses.

Related literature is poorly discussed with several omissions of related works and a rather shallow level of discussion. Highly related are the lines of work by Sisbot and Alami et al. (e.g. [2]) which is complementary to yours -- they take a model-based approach, no learning -- but are, to the knowledge of the reviewer, the first to address the issue of user preferences in robot motion planning and the work by Cakmak et al. [1] on complex manipulation tasks in an active learning framework. The line of work by Ijspeert and Schaal on DMPs seem to be underrepresented in particular given the many variations and related extensions of their original idea. The works by Calinon and Billard et al. (e.g. [3]) on probabilistic motion generalization are not at all cited and discussed. Also, it appears unjustified to dismiss the cited and highly related work [32] so easily ("since they learn in discretized action space their approach is not directly applicable to manipulators with large action spaces"). Their work was evaluated on four (continuous-state) domains similar to yours and the question of the applicability of their method should at least be elaborated in more detail.

The paper's contribution is limited by the fact that the core theoretical properties such as the TPP algorithm and the regret bounds are from previous work [28] and that the amount of manual effort in designing the features is very high (admittedly, certain context information such as object properties are hard to get automatically). This makes that the main contribution of this paper lies in the feature definition, the implementation and evaluation. Also, the paper assumes perfect knowledge of the environment and all static and dynamic objects.

Some questions remain unclear regarding interactive feedback: It is not clear what you mean by correcting waypoints, can the user just confirm/dismiss them or edit their 3D position? How many waypoints did a typical trajectory have in the experiments? How much time did the training phase take (for both kinds of feedback) and how many times bad waypoints needed to be corrected during training?

In the evaluation, the nDCG metric is not sufficiently explained neither is Table 1 (values in the cells?). The user study lacks a hypothesis and is of limited significance with N = 2.

References:
[1] M. Cakmak et al., "Designing Robot Learners that Ask Good Questions", HRI 2012
[2] E.A. Sisbot et al., "A human aware mobile robot motion planner", TRO, 2007
[3] S. Calinon et al., "Active Teaching in Robot Programming by Demonstration", RO-MAN 2007
Summary: Summarizing, a good paper with partly innovative ideas but limited contribution and strong assumptions. Weaknesses in the presentation (missing details) and evaluation are relatively easy to fix. Should the paper be rejected, I want to encourage the authors to continue this interesting work and revise their manuscript according to this feedback.

Submitted by Assigned_Reviewer_6

Update:
Thanks to the authors for their useful answer. My main concern on the feature selection still remains but from the answer I can see how the approach could generalize to similar manipulators.


Summary of the paper:
This paper presents a method to learn features for good trajectory generation for manipulators. Such problem is usually called inverse reinforcement learning (or inverse optimal control) but normally it is assumed that a set of demonstrated trajectory close to optimal is given to infer the cost function. In this paper, the only feedback comes from a user that does not need to provide optimal trajectories as example but needs to either 1) re-rank a set of trajectories proposed by the planner or 2) propose an improvement on a trajectory proposed by the planner though an interactive visualization software. The proposed algorithm as the advantage to have provable regret bounds that make it attractive to use on a real system.


Quality:
+ The proposed approach is well evaluated in the experiments show convincing results.
- It is not clear why the proposed features were selected, they seem relatively arbitrary. Why not include the manipulability measure or such measures of manipulation capabilities? Typically the trajectory features do not seem very generalizable to other robots.

Clarity:
+ The paper is very well written and well organized.
- In the experimental setup, how are the tasks of varying difficulty defined? (Sec. 5.2)


Originality:
+ In robotics, several inverse reinforcement learning algorithms have been recently proposed to infer good cost functions from demonstrations. However, in the continuous space setting, most contributions assume that the algorithm is provided with optimal demonstrations. In the present paper, this assumption is not necessary and therefore constitutes an interesting and potentially very useful contribution to the field.
+ The paper mainly re-uses a known algorithm to learn the features but the application domain is novel and the experimental results are new too.


Significance
+ A human typically does not know what combination of features will constitute a good trajectory but can tell if one motion is better than another one. From that point of view, the proposed algorithm is potentially very useful for roboticists.

Summary: The paper proposes an original approach to learn cost functions to find good motion trajectories for robot manipulators. Overall the experiments are interesting and the results potentially very useful for roboticists.

Submitted by Assigned_Reviewer_7

This paper considers the quality of instructive feedback provided to a robot as it attempts to improve its trajectory to accomplish a task. The burden of the human instructor is reduced by only requiring the demonstration of an improved trajectory rather than an optimal one.

The following comments related to particular parts of the paper.

In Section 1, third paragraph, "users ability" should be "user's ability". "improve the robots trajectories" should be "improve the robot's trajectories". In the fourth paragraph, "our algorithm learn preferences" should be "our algorithm learns preferences". Also in this paragraph, it would be helpful to fully explain what is meant by "expressive trajectory features".

At the end of Section 1, please explain what is meant by the claim that no previous work learns such preferences. Any human-directed learning process must deal with less-than-optimal examples. Stochastic gradient methods, for example, based on such feedback should learn to improve trajectories as long as parts of the instructed trajectories tend to be better.

In Section 2, second sentence, "mimicking experts" should be "mimicking experts'" In last sentence of Section 2, "humans-robot interaction" should be "human-robot interaction".

In Section 3, Step 2, it is not clear how the user "sees" the trajectory before it is executed. Introducing Figure 1 before this would help reader see that potential trajectories can be shown on the display. Also, Step 2 mentions the requirement that the trajectory provided by the user is superior to the robot's top-ranked trajectory. What if this is not true? There must be numerous situations where the user might provide an inferior trajectory due to lack of consideration of all aspects of the trajectory and its relation to objects.

In Section 4.2, you say each trajectory is divided into three parts, analyzed separately. Please explain why three parts are sufficient?
Summary: This paper provides an interesting approach to learning improved trajectories from user feedback indicating trajectories that are better, but perhaps not optimally better, than the current trajectory. Many experiments with users are presented; fewer experiments with more in-depth analysis of the results would make this paper more comprehensible.
Author Feedback

Author rebuttal: R6:

Thank you for your review, and finding our `application domain novel’ and our contribution on not needing `optimal demonstrations’ in `continuous space setting’ an `interesting and potentially very useful contribution to the field’.

R6 asked for generalizability of our approach to different robots. We appreciate R6’s suggestions on additional features such as manipulability measure, and we will consider adding them. In response, our algorithm/features can easily be adapted for different robots. In order to demonstrate this, we recently implemented it on the Baxter robot (in addition to PR2) and trained it to perform grocery store checkout tasks (using same algorithm and features as in the submission). An anonymous* video showing this is available at: [5] (below)
*Modified

R7:

R7 asks what if the user “provides an inferior trajectory as feedback”. Our TPP algorithm only requires the feedback to be alpha-informative in _expectation_ and not deterministically (see Sec 4.4).

We thank R7 for his careful review and pointing out grammatical issues. We’ll fix them in the final version. The sentence “no previous work learns such fine-grained preferences ..” emphasizes that in unstructured environments such as homes, context with objects plays a crucial role and preferences vary with users, task and environment. (Previous works in IRL such as autonomous helicopter flight, ball-in-a-cup manipulator etc., focus on specific motions or paths but context is of limited importance.) We’ll edit the text to clarify.


R1:

Thank you for your detailed review, specific questions, and suggested references to additionally cite. We’ll edit text to clarify. Below are a few clarifications.

** Size of user study

The rationale behind user study is to show that eliciting suboptimal feedback for high DoF robots is indeed possible and practically realizable. Using the same algorithm and features as in the submission, we now have a larger user study on a second robot ([5]), which shows similar trends and conclusions.

** Contribution over previous work:

We’d like to point that we go substantially beyond existing work in the following three ways:

1) First, we extend coactive learning from discrete and rather simple objects, like document ranking [28], to predicting highly complex and high dimensional continuous objects like trajectories while retaining its theoretical guarantees.

2) Second, we go beyond the previous work on trajectories by relating preferences with object affordances, e.g., heavy objects have different affordances than fragile items. This allow us to handle applications where the context is important such as robots for household chores, assembly lines, grocery store checkout, etc.

3) Third, we only require users to provide suboptimal suggestions for improvement and do not depend on their ability to provide (near) optimal demonstrations. Our approach also provides a model for suboptimal user feedback and under which it converges to user’s hidden score function. We find this aspect missing in previous literature on IRL and incremental learning (R1 reference [3], [4] (below) and [32]).

** Other references

We thank R1 for pointing additional relevant papers and we’ll cite them in the final version. We highlight the differences from these papers below:

The application we address puts us in contrast to R1 references. In detail, we differ from Calinon et. al., [4] and R1 reference [3], on the following:
1. They deal with relatively simpler tasks in obstacle free environment (e.g. moving chess pieces), while our environments/dataset contain many objects.
2. Incremental learning [4] does not model the suboptimality in user demonstration and nor it provides theoretical guarantees. We model user feedback as alpha-informative and our algorithm converges (theoretically proven) to user’s hidden score function.
3. They find an optimal controller to mimic user. On the other hand, we learn score function, which allows ranking trajectories and provide fallback options when a trajectory fails.

As for Camak et. al. and Sisbot et. al. (R1 reference [1] and [2]), it is easy to distinguish our work. Sisbot et. al. follow a model based approach to construct cost functions for simple 2D paths, while we `learn’ them for high dimensional trajectories. Camak et. al. provides an interesting framework wherein the robot ask questions to elicit demonstrations. However, while still relying on user’s ability to demonstrate trajectories, they use pre-scripted set of questions and do not provide methods for active query.


Other Minor Questions

[R6] “In the experimental setup, how are the tasks of varying difficulty defined?”

The experimental setup does not require knowledge of task difficulty. However, for the presentation of the results, we ordered them informally based on expert’s opinions.

[R7] The rationale behind dividing trajectory in three parts is that trajectories include a beginning and end interaction with the environment which can be critical for certain tasks such as pouring and pick-and-place. However, other choices of features could be investigated as well.

[R1] “what you mean by correcting waypoints?”

It means changing robot’s arm configuration to desired position at a particular time-instant.

[R1] `the number of waypoint?’

It depends on trajectory length, but our algorithm and results do not depend on their number. Since trajectories are continuous, user can change robot’s configuration anywhere on the trajectory.

[R1] `How much time did the training phase take?’

In our larger user study, on average a user took 5.5 min, 3 re-rank and 2 interactive feedback per-task (total of 10 tasks). We’ll include these results and related discussion in the final version.

Reference:
[4] S. Calinon et. al., Incremental Learning of Gestures by Imitation in a Humanoid Robot, HRI 2007
[5] http://pr.cs.cornell.edu/coactive